# A Comparison of Wave Spectra during Pre-Monsoon and Post-Monsoon Tropical Cyclones under an Intense Positive IOD Year 2019

**Mourani Sinha** [1],*[iD], **Somnath Jha** [2] **and Anupam Kumar** [3]

1   Department of Mathematics, Techno India University, Kolkata 700091, West Bengal, India
2   West Bengal Police Service, Home & Hill Affairs Department, Balurghat 733101, West Bengal, India
3   Centre for Climate Research Singapore, Meteorological Services Singapore, National Environment Agency, Singapore 228231, Singapore
*   Correspondence: mou510@gmail.com; Tel.: +91-9350904194

**Abstract:** The impact of Indian Ocean Dipole (IOD) events on the generation and intensity of tropical cyclones under the influence of monsoons is explored. The standardized sea surface temperature (SST) anomalies are computed for the pre-monsoon and post-monsoon months for the Bay of Bengal (BOB) and Arabian Sea (AS) from 1971 to 2020 and relationships are analyzed with the frequency of tropical cyclones for the neutral, positive and negative IOD years. Ocean states are sensitive to cyclonic conditions exhibiting a complex spectral distribution of the wave energy. Due to a tropical cyclone, the surface waves remain under high wind forcing conditions for prolonged periods generating a huge amount of energy. The spectral wave model SWAN (Simulating WAves Nearshore) is used to generate the energy density spectra during FANI (26 April–5 May 2019), which was a pre-monsoon extreme severe cyclonic storm, and BULBUL (5–12 November 2019), which was a post-monsoon very severe cyclonic storm in the BOB region. This study aims to estimate the intensity of wave energy during tropical cyclones in the pre- and post-monsoon months for 2019 (an extremely positive IOD year).

**Keywords:** Indian Ocean Dipole; tropical cyclones; energy density spectra; numerical wave model; monsoon; sea surface temperature





## 1. Introduction

The tropical Indian Ocean is impacted by the Indian Ocean Dipole (IOD), which is a coupled air–sea dipole mode influencing a large number of lives along the coast. The extreme positive IOD years have seen floods in the eastern African countries and droughts in the countries along the eastern Indian Ocean [1]. The extreme positive IOD event in 2019 was the strongest prolonged one since the 1960s [2–5] and it led to basin-wide warming in the tropical Indian Ocean in 2020 [6]. In another study, [7] showed how the IOD events influenced the tropical cyclone variability in the Western North Pacific Ocean. In this paper, the frequency of cyclones during pre- and post-monsoon periods in the Arabian Sea (AS) and Bay of Bengal (BOB) has been analyzed for the IOD years from 1971 to 2020. Then, considering an intense positive IOD year (2019), the spectral wave energy is estimated during pre-monsoon and post-monsoon tropical cyclones. [8] showed negative IOD years to be more suitable for formation of intense cyclones in the BOB during October–December. [9] analyzed BOB-OND cyclone frequency for the period 1979–2020 and showed an increased number of cyclones during the negative IOD years compared to the neutral and positive phases. In this study, for the period 1971 to 2020, similar results were obtained. The novelty of the study lies in the analysis of extreme cyclones formed during an intense positive IOD year (2019).

The oceanic general circulation is regulated by the mechanical energy produced due to wind stress, tidal dissipation and tropical cyclones [10]. In that study, they calculated

the mechanical energy input to the global oceans due to tropical cyclones from 1984 to 2003 running a hurricane–ocean coupled model. Due to tropical cyclones, they estimated the annual mean energy input is 1.62 TW considering the surface waves and 0.10 TW for the surface currents. [11] reconciled and compared various interpretations of kinetic energy generation. They revisited various terms of the global kinetic energy budget in the context of a tropical cyclone and discussed the kinetic energy equation given by [12] and [13]. Due to tropical cyclones, huge amounts of energy are released into the upper ocean. This increases the vertical mixing and thus lowers the sea surface temperature. [14] studied the impact of additional vertical mixing due to unbroken surface waves during tropical cyclones. They tested a new parameterization of surface wave mixing related to tropical cyclone passage and the change in the total kinetic energy production. [15] reviewed tropical cyclone wind and wave fields and commented that ocean wave models need to modify the nonlinear source term. They reported that tropical cyclone spectra can be uni-modal or bi-modal, having separate wind and swell components [16–21].

The energy density spectrum represents the distribution of wave energy at different wave frequencies. On availability of directional information, a directional ocean wave spectrum is demonstrated. A sea-state can be quantified by plotting the energy density as a function of frequency. On 24 August 1998, NASA and NOAA together measured for the first time the sea surface directional wave spectrum of hurricane Bonnie that made landfall in North Carolina, USA [18]. The NOAA hurricane research aircraft carrying the NASA scanning radar altimeter provided measurements in all the four quadrants of the inner core of the hurricane. The highest waves were observed in the right forward quadrant and the lowest waves in the left rear quadrant of the hurricane. In another study, the directional wave spectrum of the same hurricane Bonnie was estimated [20] using ocean surface wave model WAVEWATCH III [22]. In the open ocean, model-simulated and NASA Scanning Radar Altimeter observed spectra matched in all the four quadrants of the hurricane. Further wave spectra data from 16 hurricanes off the northwest coast of Australia from Datawell Waverider Buoys over a 16-year period were examined [16]. The parametric form of the hurricane-generated wave spectra was determined. The 229 uni-modal spectra examined were those that were related to hurricanes having central pressures from 905 to 985 hPa. There was no conclusion about the directional properties of the spectra in this study. Ref. [17] further examined the directional wave spectrum of nine tropical cyclones along Australia's northwest coast during 1995–2000, mostly using directional wave buoy data and observed swell-dominated spectra.

In the Gulf of Mexico, directional wave spectra were studied from data collected hourly from 12 buoys for seven hurricanes occurring [21]. The hurricane wave spectra were mostly swell-dominated and uni-modal, similar to the fetch limited cases. [23] continued the analysis of directional wave spectra inside hurricanes. They represented the azimuthal and radial variations of the propagating wave directions and also presented a wave propagation model giving the dominant wave direction. Except for a landfalling case and a fast-changing hurricane, the buoy-measured and model-simulated wave directions were in agreement.

The most common environmental impact of high-density wave energy is coastal erosion. This affects the life and property of all those living near the coast. To be mentioned, some of the marine renewable energy sources are tides, ocean currents, waves, offshore wind, thermal differences and salinity gradients. Of these, the wave energy has the maximum potential in terms of density. Wave energy has become the most reliable renewable source of marine energy to meet electricity demand and reduce greenhouse gas emissions. Thus, the estimation of wave energy during tropical cyclones is a major area to be explored. [24], using the SWAN model [25], evaluated the wave energy potential in the southern Caspian Sea. They assessed that the stations located near the coast are the most appropriate ones for wave energy harvesting. The model was simulated for 11 years and the central part of the southern Caspian Sea was seen as the highest wave power generating region. In another study, the storm-generated wave fields were compared with the buoy observations for the Canadian Beaufort Sea [26]. In this study, the SWAN model is used to

generate the one-dimensional energy density spectrum in the BOB region and the intensity of the wave spectra are estimated during pre- and post-monsoon cyclonic periods. The novelty of the study lies in the fact that the wave spectra analysis is conducted during an intense positive IOD year (2019) and its impact is discussed. For a span of 50 years (1971–2020), the number of cyclones occurring were analyzed for the BOB and AS regions for the neutral, positive and negative IOD years. FANI and BULBUL were two severe cyclonic storms in the BOB that occurred in the pre- and post-monsoon months, respectively, in 2019 (an intense positive IOD year), and thus are chosen for wave spectra analysis.

## 2. Data and Methodology

### 2.1. Data

The SST data were downloaded (https://psl.noaa.gov/data/gridded/data.cobe.html (accessed on 29 December 2022)) for the period from 1971 to 2020 (50 years) and averaged for pre-monsoon months comprising March, April and May (MAM) and post-monsoon months comprising October, November and December (OND) for the BOB and AS separately. The number of cyclones was analyzed for BOB−MAM, BOB−OND, AS−MAM and AS−OND using IMD cyclone e−atlas (https://rsmcnewdelhi.imd.gov.in/ (accessed on 29 December 2022)). The standardized SST anomalies were computed for the BOB−MAM, BOB−OND, AS−MAM and AS−OND space time for 50 years and relationships were explored with the frequency of cyclones during the neutral, positive and negative IOD years (Figure 1). Monthly dipole mode index data were obtained (https://psl.noaa.gov/gcos_wgsp/Timeseries/DMI/ (accessed on 29 December 2022)) having neutral bounds between −0.4 °C and +0.4 °C. To calculate the standardized anomalies, all the years from 1971 to 2020 were considered as the climatological base period.

**Figure 1.** IOD index (degC) for all the months from 1971 to 2020.

### 2.2. Wave Energies

Next, to generate the wave energies, the SWAN model was integrated for the Indian Ocean region extending from 30° E to 120° E longitudes and 30° N to 70° S latitudes. Bathymetry was generated using ETOPO1. SCATSAT−1 daily wind data having a 25 by 25 km spatial resolution were used as input to the model. Two cyclones were chosen, namely, FANI (26 April–5 May 2019) during MAM, which was an extremely severe cyclonic storm, and BULBUL (5–12 November 2019) during OND, which was a very severe cyclonic storm, in the BOB region to estimate the energy density spectra. The model was simulated for April–May 2019 and October–November 2019 having 25 km spatial resolution. The model was run in two-dimensional non-stationary mode and wave spectra were generated at grid points (84° E, 15° N) and (85° E, 18° N) for wave energy estimation during the FANI cyclone. Similarly, for the BULBUL cyclone, grid points (87.5° E, 20° N) and (88° E, 21° N) were chosen for wave spectral generation.

*2.3. SWAN Model and SCATSAT-1 Product*

The third-generation spectral wind-wave models such as WAM [27], WAVEWATCH III [22] and SWAN [25] solve the spectral action balance equation without any restrictions on the shape of the spectrum. The shallow-water wave model SWAN incorporates the deep-water features such as wave generation, dissipation and the quadruplet wave–wave interactions from the WAM model [28]. The shallow-water features include dissipation due to bottom friction, triad wave–wave interactions and depth-induced breaking.

Ocean surface winds are useful for severe weather studies and numerical weather prediction models. SCATSAT-1 was built and launched by the Indian Space Research Organization on 26 September 2016 from Satish Dhawan Space Centre, Sriharikota [29] discussed applications of SCATSAT-1 spatial wind vector fields over the northern Indian Ocean for tropical cyclone prediction. From the India Meteorological Department, the best track of six tropical cyclones was obtained and studied. SCATSAT-1 data (https://www.mosdac.gov.in/ (accessed on 20 November 2019) were utilized for tropical cyclogenesis prediction, center identification, size determination, wind asymmetries analysis, etc. [30] discussed the comparison of the SCATSAT-1 data with other scatterometer and numerical model data. Scatterometer onboard satellites are instruments that measure wind speed and wind direction over the surface of the ocean. Scatterometers are microwave radar systems that first send microwave pulses to the earth's surface and then measure the backscattered power.

In this study using the SWAN model, the wave power intensity was estimated during tropical cyclones in the BOB region. Two cyclones occurring in the pre-monsoon and post-monsoon seasons in the BOB were considered for the study. FANI was an extreme severe cyclonic storm causing extensive damage along the Odisha coast during April–May 2019 and BULBUL was a very severe cyclonic storm making landfall near the West Bengal and Bangladesh coast during November 2019. The spectral wave model SWAN was integrated to generate the cyclonic wave fields and the energy density spectra at particular grids. For accuracy, scatterometer SCATSAT-1 (25 km) daily wind fields given by the Space Application Centre of the Indian Space Research Organization were given as input to the model. The characteristics of the spectra were compared in this study during the pre-monsoon and post-monsoon seasons for an intense positive IOD year, 2019.

Figure 2 shows a comparison between buoy-observed and SWAN-model-computed daily averaged significant wave height (SWH) values at Gopalpur (85° E, 19.25° N) during the cyclone FANI. The buoy SWH data were obtained from INCOIS (Indian National Centre for Ocean Information Services, Hyderabad, India) at 30 min intervals and averaged daily for comparison with SWAN model computations. Although there was under-prediction due to the input winds, the trend of the observed and simulated wave heights matched well.

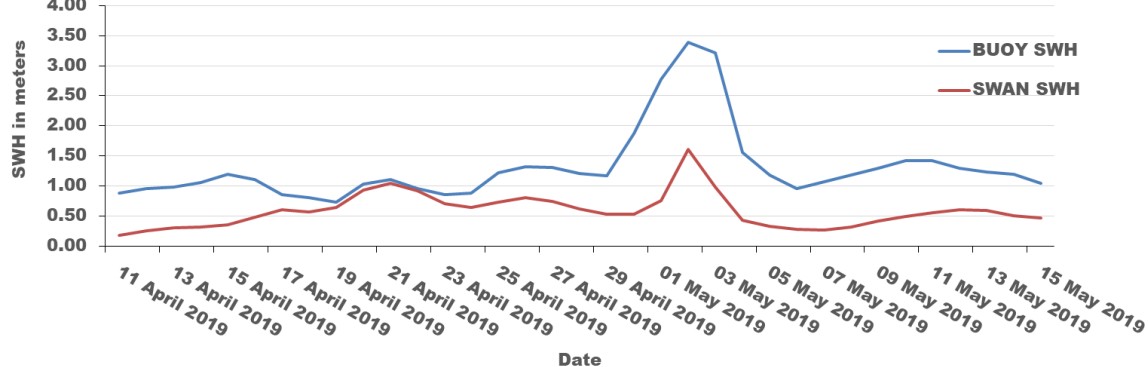

**Figure 2.** Comparison of observed and computed daily averaged SWH at Gopalpur (85° E, 19.25° N) during the cyclone FANI.

## 3. Results and Discussion

Two regions containing the eastern and western IOD regions were chosen in the Indian Ocean, namely, the BOB (78° E–110° E and 25° N–10° S) and AS (50° E–78° E and 25° N–10° S), and SST data were averaged for MAM and OND from 1971 to 2020. The number of cyclones was analyzed for BOB-MAM, BOB-OND, AS-MAM and AS-OND, and the corresponding standardized SST anomalies were computed. Scatter plots were generated considering all the years together from 1971 to 2020 (Figure 3), only the positive IOD years (Figure 4), only the negative IOD years (Figure 5) and the neutral IOD years (Figure 6). From the analysis, it could be said that there is no significant relationship between the standardized SST anomalies of any region with its pre- or post-monsoon cyclone frequency except for post-monsoon cyclone frequency in the Arabian Sea during positive IOD years (Figure 4). Thus, the positive standardized anomaly of the Arabian Sea has driven more cyclone frequency. As given by IMD cyclone e-atlas (https://rsmcnewdelhi.imd.gov.in/ (accessed on 29 December 2022)), the number of cyclonic storms of different categories is denoted by CY.

The numbers of cyclones were grouped into intervals of five years for the period 1971–2020 (Table 1) for BOB-MAM, BOB-OND, AS-MAM and AS-OND. Figures 7 and 8 show that the frequency of cyclones during pre- and post-monsoon periods in the BOB and AS has quite different trends. The trend is not so significant in the AS, but in the BOB, there is a decreasing trend during the post-monsoon period. In the BOB, a conspicuous scenario could be detected that there is no significant trend of cyclones during the pre-monsoon period but a significantly ($R^2 = 0.7363$) decreasing trend of cyclone frequency could be detected during the post-monsoon period (Figure 7). Thus, observational analysis of this present study indicates that there is a significantly decreasing trend of cyclones during the post-monsoon period in the BOB in the five-year trend, whereas the warming of the AS during the post-monsoon period during positive IOD years indicates an increased number of cyclones in the AS.

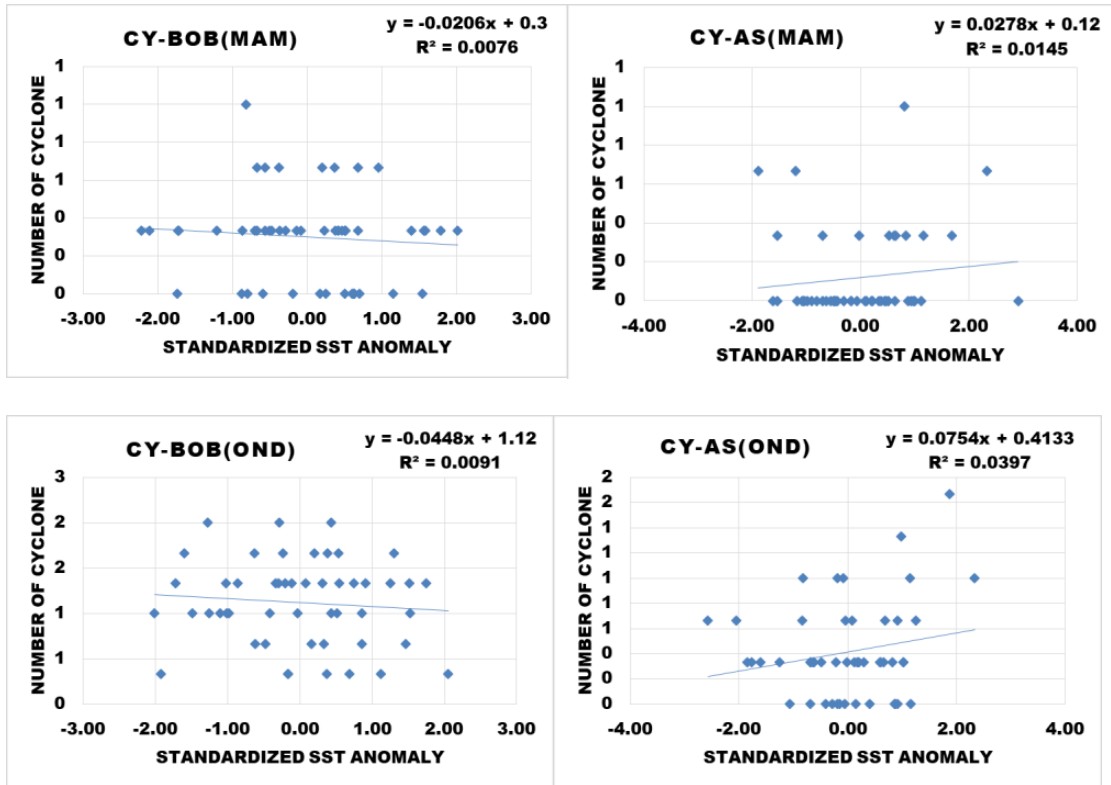

**Figure 3.** Scatter plots between standardized SST anomalies and number of cyclones (CY) for all the years (1971–2020) together for BOB−MAM, AS−MAM, BOB−OND and AS−OND.

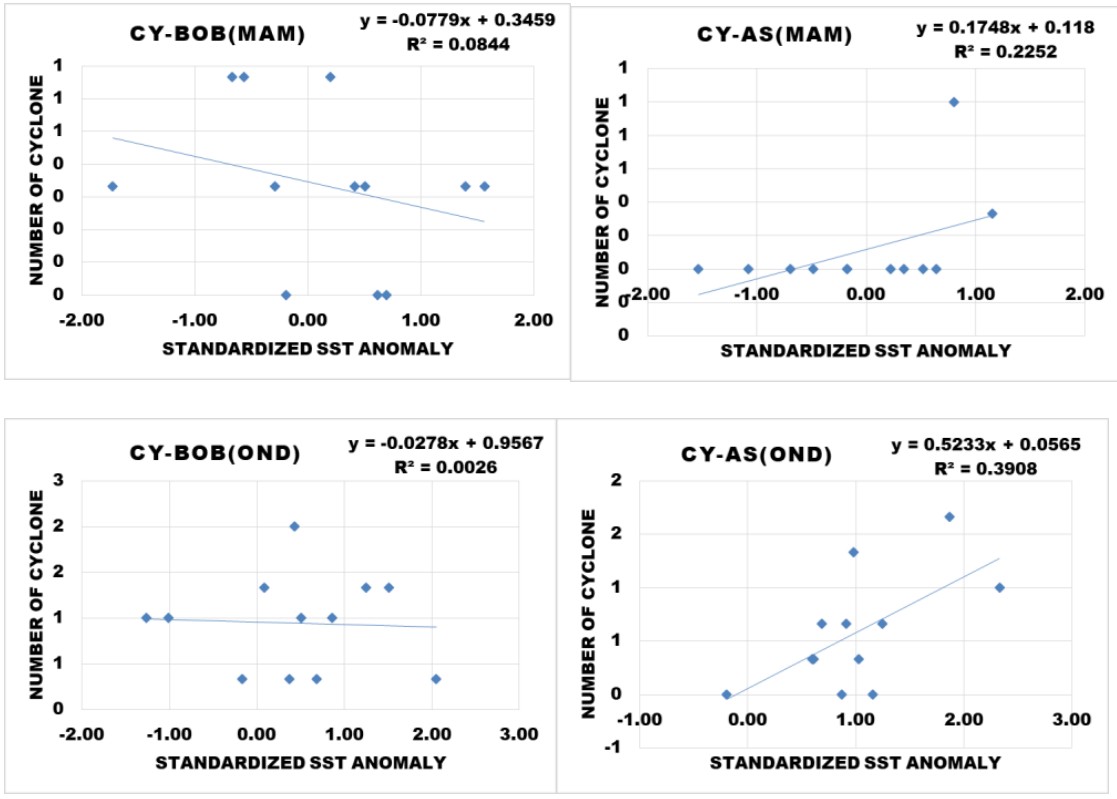

**Figure 4.** Scatter plots between standardized SST anomalies and number of cyclones (CY) for the positive IOD years for BOB−MAM, AS−MAM, BOB−OND and AS−OND.

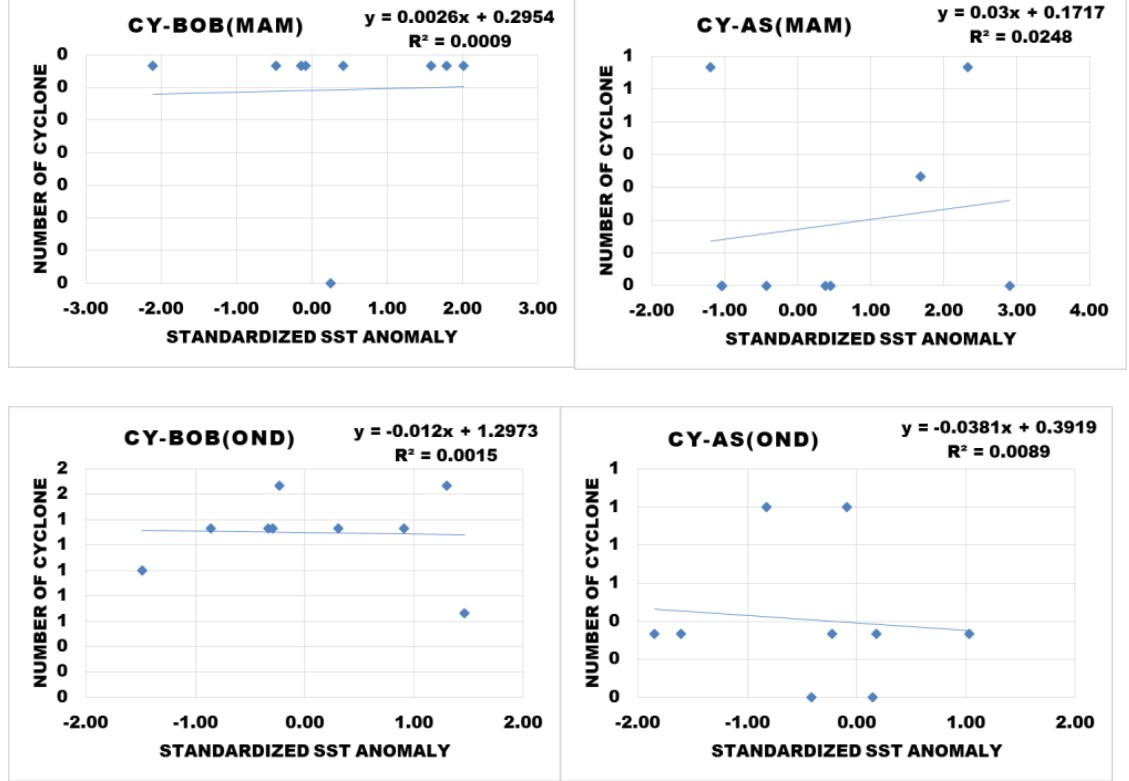

**Figure 5.** Scatter plots between standardized SST anomalies and number of cyclones (CY) for the negative IOD years for BOB−MAM, AS−MAM, BOB−OND and AS−OND.

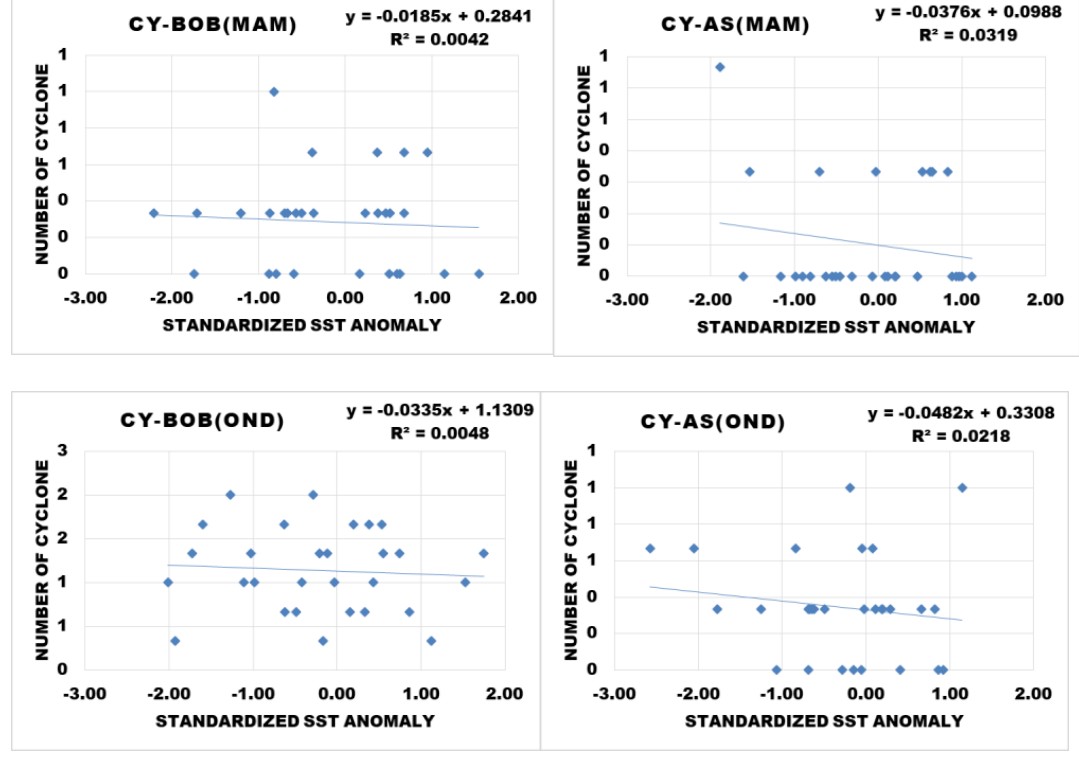

**Figure 6.** Scatter plots between standardized SST anomalies and number of cyclones (CY) for the neutral IOD years for BOB−MAM, AS−MAM, BOB−OND and AS−OND.

**Table 1.** Number of cyclones from 1971–2020 at intervals of 5 years.

| Year Range | CY-BOB (MAM) | CY-BOB (OND) | CY-AS (MAM) | CY-AS (OND) |
|---|---|---|---|---|
| 1971–1975 | 1 | 8 | 2 | 3 |
| 1976–1980 | 2 | 6 | 0 | 3 |
| 1981–1985 | 1 | 7 | 0 | 2 |
| 1986–1990 | 1 | 6 | 0 | 1 |
| 1991–1995 | 3 | 5 | 0 | 2 |
| 1996–2000 | 1 | 5 | 1 | 2 |
| 2001–2005 | 1 | 5 | 1 | 1 |
| 2006–2010 | 2 | 4 | 1 | 1 |
| 2011–2015 | 1 | 4 | 0 | 3 |
| 2016–2020 | 2 | 5 | 1 | 3 |
| Total Cyclones | 15 | 56 | 6 | 21 |

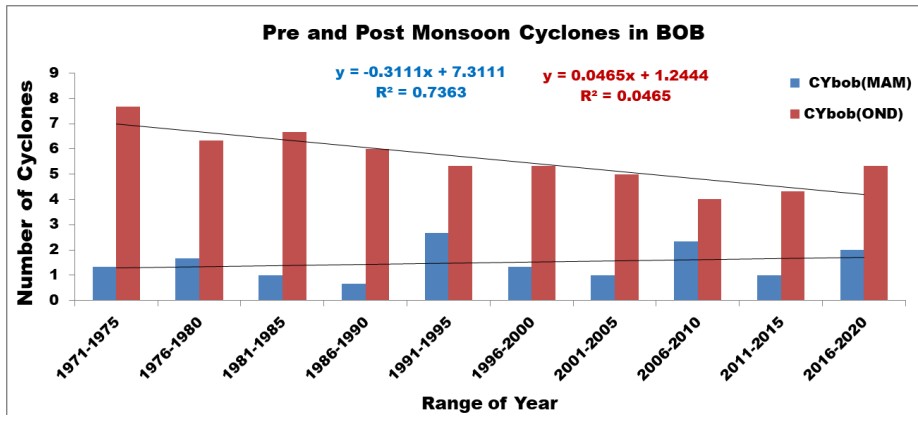

**Figure 7.** Comparison of pre−monsoon and post−monsoon cyclone frequencies in BOB.

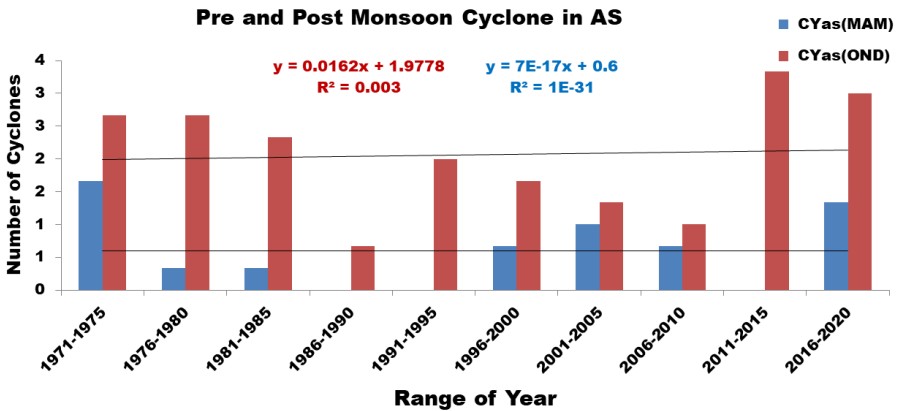

**Figure 8.** Comparison of pre−monsoon and post−monsoon cyclone frequencies in AS.

Further analysis of the pre−monsoon and post−monsoon cyclones for the BOB and AS, IOD−year−wise (negative, neutral, positive), reveals that the maximum cyclones occurred during neutral years followed by positive IOD years in the post−monsoon season (Table 2 and Figure 9).

**Table 2.** Number of cyclones from 1971 to 2020 for negative, neutral and positive IOD years.

| IOD Years (50) | CY-BOB (MAM) | CY-BOB (OND) | CY-AS (MAM) | CY-AS (OND) | Total |
|---|---|---|---|---|---|
| Negative (9) | 3 | 12 | 2 | 4 | 21 |
| Neutral (29) | 8 | 33 | 3 | 10 | 54 |
| Positive (12) | 4 | 11 | 1 | 7 | 23 |
| Total | 15 | 56 | 6 | 21 | |

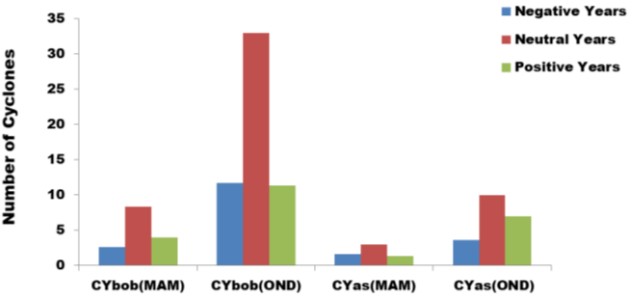

**Figure 9.** Frequencies of pre−monsoon and post−monsoon tropical storms during IOD years (1971–2020) in BOB and AS.

Next, severe tropical cyclones (FANI and BULBUL) were chosen that occurred in the BOB during pre−monsoon and post−monsoon periods for an intense positive IOD year (2019) and the wave spectra were analyzed.

The SWAN model was simulated to generate the wave spectra at particular grid points in the BOB region through which the FANI and BULBUL cyclones passed in 2019. Daily wind fields given by SCATSAT−1 scatterometer were used as an input to the model. In the first experiment, 25 km gridded daily wind fields were used and wave fields were generated for the entire Indian Ocean during the period of FANI and BULBUL. The model was run for April and May 2019 to simulate the wave fields and spectra during FANI (26 April–5 May 2019), which was an extreme severe cyclonic storm over the BOB. Figure 10 depicts the spatial wind vector fields from 28 April to 3 May 2019 of the FANI cyclone, given by SCATSAT−1 (25 km). The scatterometer winds clearly depict the cyclonic wind pattern.

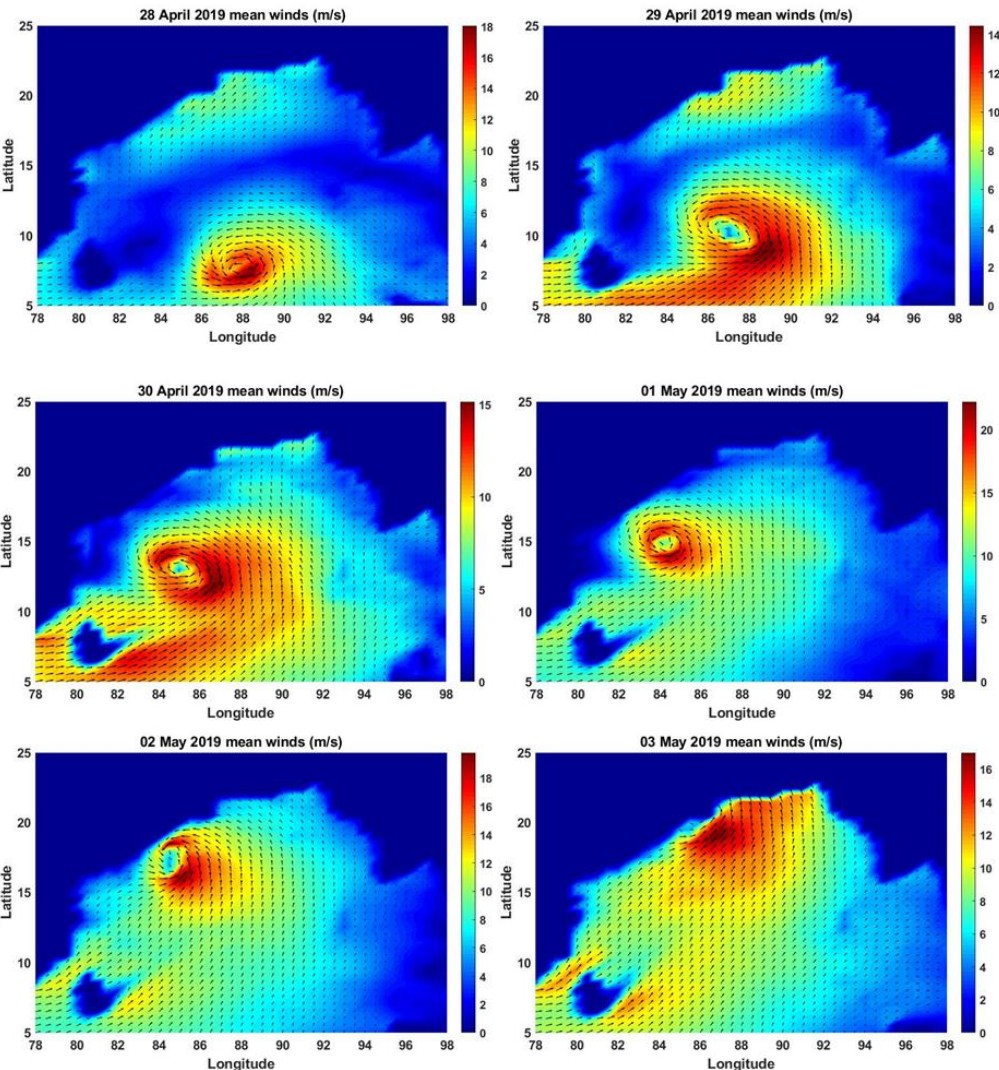

**Figure 10.** Mean winds (m/s) during FANI cyclone (28 April to 3 May 2019) from SCATSAT−1 (25 km) daily products.

The corresponding daily significant wave height fields were generated during the FANI cyclone using the SWAN model. Figure 11 depicts the wave height fields during 1–3 May. On 3 May, the cyclone made landfall off Odisha coast. Daily maximum wave heights of more than 3 m on 2 May and more than 2 m on 3 May were observed.

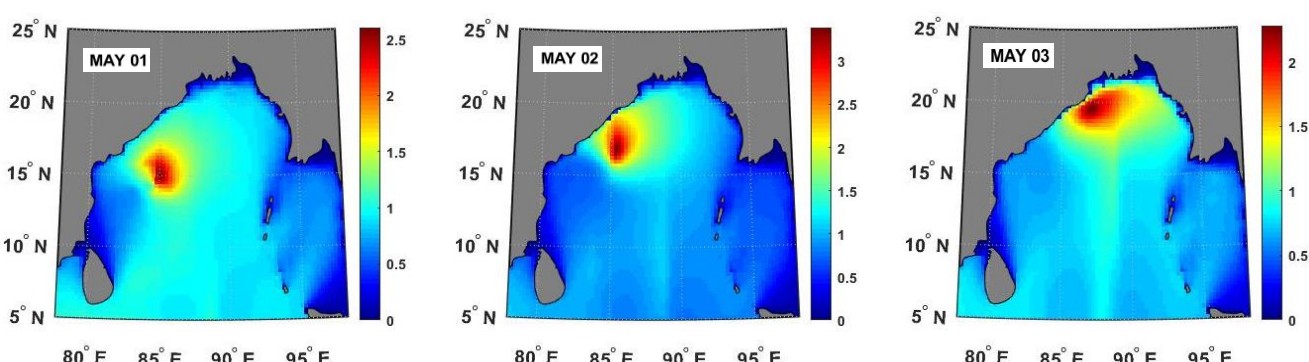

**Figure 11.** Computed SWH (m) for FANI cyclone from 1 to 3 May 2019 using SCATSAT-1 daily 25 km wind data.

Along the east coast, daily wind speeds reached more than 20 m/s or 72 kmph on 1 and 2 May. The computations were under-predicted due to rain contamination in the scatterometer data and the daily averaged value. On 2 May, the cyclone passed the grid point (84° E, 15° N) as an extreme severe cyclonic storm. Figure 12 gives the spectral distribution of the daily wave spectra from 28 April to 3 May 2019 at the grid point mentioned.

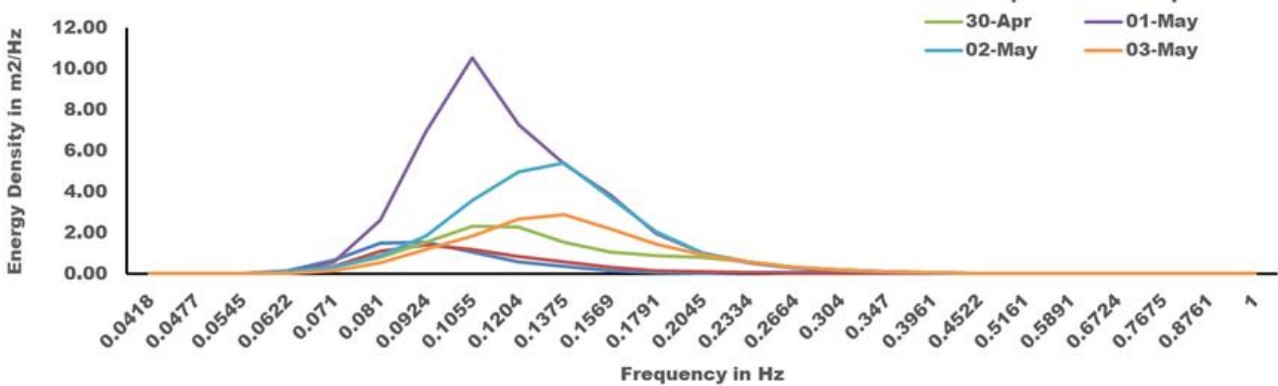

**Figure 12.** Computed daily wave spectra ($m^2$/Hz) for FANI cyclone at 84° E and 15° N using SCATSAT-1 daily 25 km wind data.

The energy density peaked on 1 May followed by 2 May. The second grid was chosen positioned to the right and northwards of the first grid. The cyclone FANI passed as an extremely severe cyclonic storm through the second grid (85° E, 18° N) also on 2 May. Figure 13 shows the energy density spectra from 28 April to 3 May 2019.

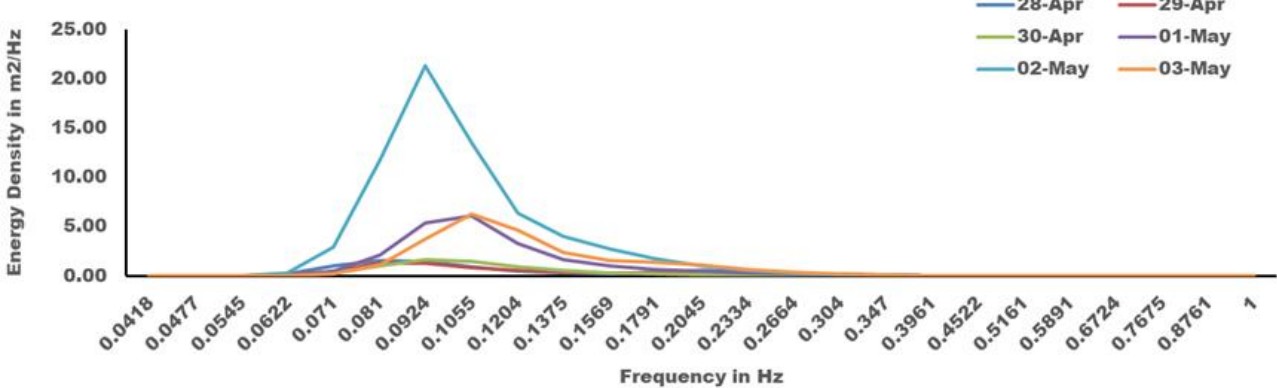

**Figure 13.** Computed daily wave spectra ($m^2$/Hz) for FANI cyclone at 85° E and 18° N using SCATSAT-1 daily 25 km wind data.

In this case, the maximum energy occurred on 2 May when the cyclone was extremely severe at the location indicated. This is in agreement with the previous studies stating the highest waves were found in the right forward quadrant of the cyclone (Wright et al., 2001). The shape of the spectra for both the grid points were uni-modal at the peak times. To simulate the wave fields during BULBUL (5–12 November 2019), the SWAN model was again integrated for the entire Indian Ocean for October and November 2019. Figure 14 illustrates the wind vector fields of the cyclone BULBUL from its generation to dissipation using SCATSAT-1 retrievals. The maximum wind speed was more than 25 km per hour (low values due to rain contamination) on 9 November near to the landfall in the head of the BOB.

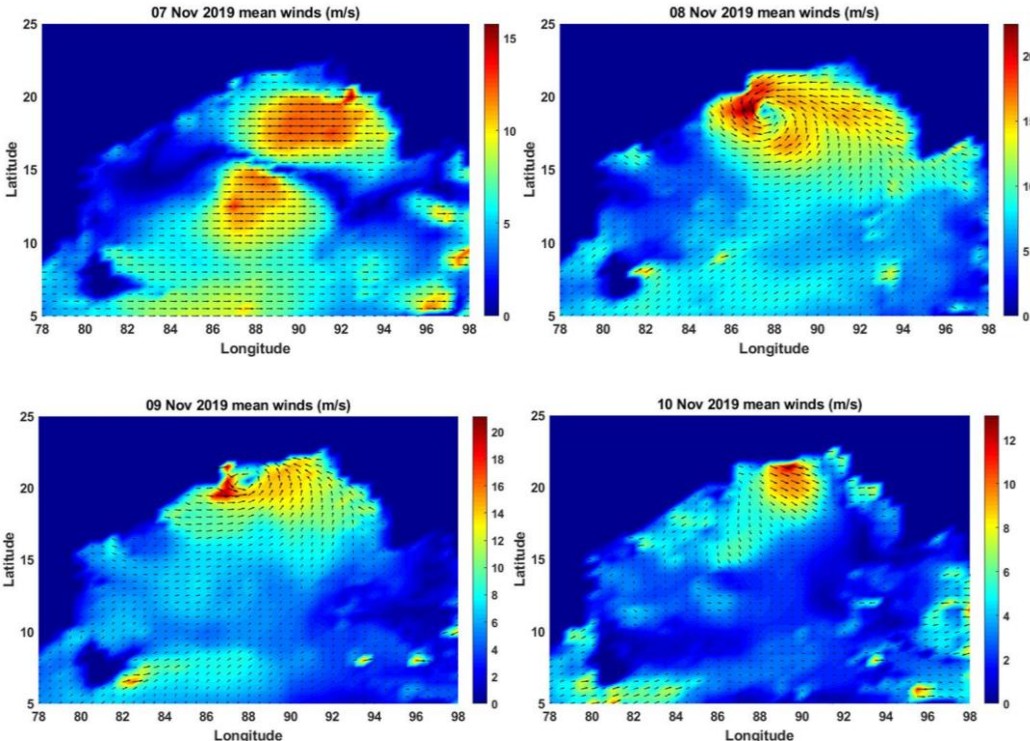

**Figure 14.** Mean winds (m/s) during BULBUL cyclone (7–10 November 2019) from SCATSAT−1 (25 km) daily products.

The cyclonic wave fields generated by the SWAN model during the BULBUL cyclone are shown in Figure 15. As shown in Figure 15, the significant wave heights, although not very large in magnitude, show increased value along the track of the cyclone on 8 and 9 November. The spatial distribution of the wave fields is in agreement with the storm conditions. Energy density spectra were generated using the SWAN model at the locations (87.5° E, 20° N) and (88° E, 21°N) from 5 to 12 November. On 9 November at 6 h, the BULBUL cyclone passed the first location as a very severe cyclonic storm in the BOB region. The cyclone passed the second location, which is to the right above, as a severe cyclonic storm on 9 November at 12 h. Figure 16 demonstrates the spectral energy density from 7 to 9 November at the location (87.5° E, 20° N). Maximum energy was observed on 8 and 9 November along with uni−modal shape. Figure 17 shows the daily wave spectra at the location (88° E, 21° N) for the BULBUL cyclone. The highest waves were observed again on 8 and 9 November but with reduced magnitude. The cyclone passed as a very severe cyclonic storm through the first location and as a severe cyclonic storm through the second location. Although the magnitude of the wave energy generated reduced, there was no significant change in the spectral shape.

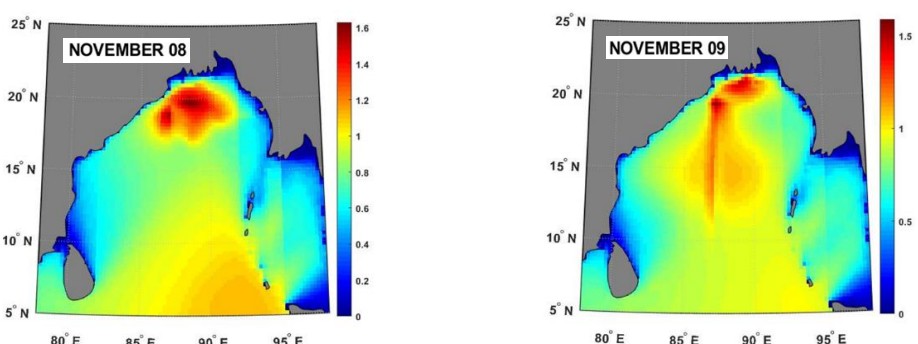

**Figure 15.** Computed SWH (m) for BULBUL cyclone on 8 and 9 November using SCATSAT−1 daily 25 km wind data.

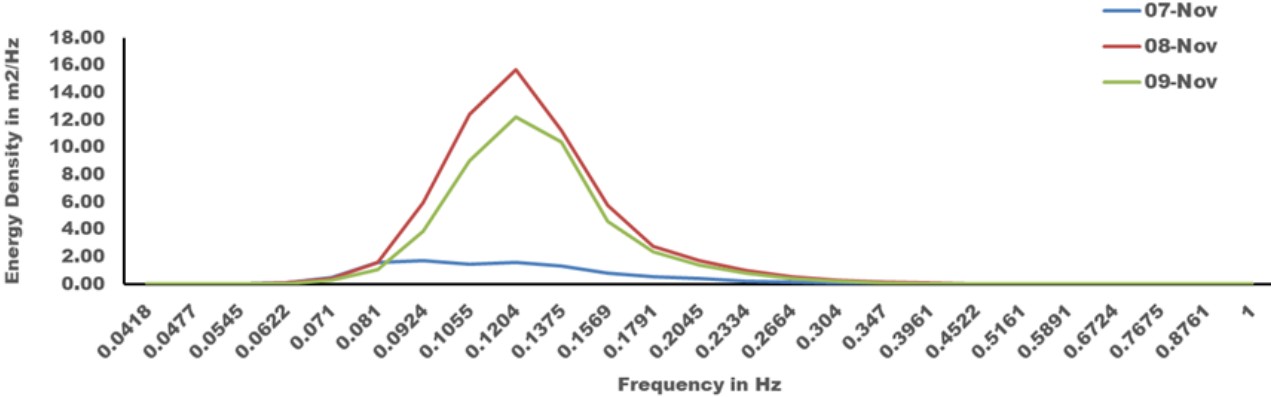

**Figure 16.** Computed daily wave spectra (m$^2$/Hz) for BULBUL cyclone at 87.5° E and 20° N using SCATSAT−1 daily 25 km wind data.

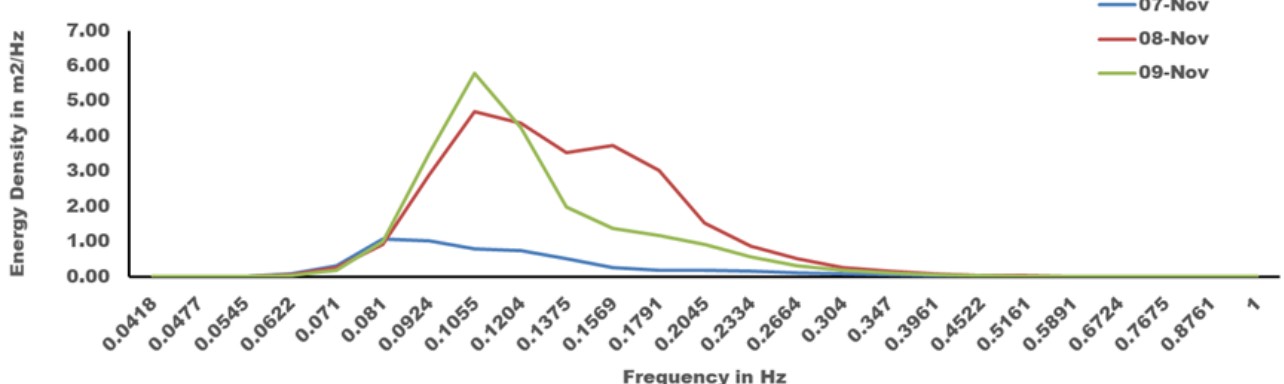

**Figure 17.** Computed daily wave spectra (m$^2$/Hz) for BULBUL cyclone at 88° E and 21° N using SCATSAT−1 daily 25 km wind data.

The intensity of the cyclone FANI was more than BULBUL, although it occurred during pre-monsoon months. As expected, the wave energies were higher during FANI compared to BULBUL, but for the latter, higher energies could be observed for a longer period.

## 4. Conclusions

The BOB region experiences cyclones both in the pre-monsoon and post-monsoon seasons and the coastal regions are vulnerable to inundations due to high storm surges. High waves cause extensive damages to buildings and constructions. Climate change and IOD phases have enhanced these risks manifold with increasing uncertainties and the advent of a new changing trend of cyclone frequencies in the AS and BOB. The present study has been conducted to assess the pre- and post-monsoon cyclone frequency of the AS and BOB and the observational data of cyclone frequencies in the above two basins and the SSTs have been analyzed. The trend of five-year group statistics has been analyzed to detect the neo-normal of the data. The analysis has revealed that no significant trend in pre- and post-monsoon cyclones could be found in the AS in the five-year trend analysis. In the BOB, a conspicuous scenario could be detected that there is no significant trend of cyclones during the pre-monsoon period but a significantly decreasing trend of cyclone frequency could be detected during the post-monsoon period. Thus, observational analysis of this present study indicates that there is a significantly decreasing trend of cyclones during the post-monsoon period in the BOB in the five-year trend, whereas the warming of the AS during the post-monsoon period during positive IOD years indicates an increased number of cyclones in the AS.

The next part of the study of wave energy is of great importance to mitigate the effect of natural hazards along the coast and also to all those venturing into the deep

oceans. Wave energy can be considered a dependable source of renewable marine energy to meet electricity demand and reduce greenhouse gas emissions. For the period of 50 years (1971–2020), the standardized SST anomalies were computed for the BOB and AS regions for the pre- and post-monsoon months. The number of cyclones during positive IOD years was found to be in relationship with the SST anomalies in the AS. Again, in the BOB, there is a decreasing trend of the number of cyclones in the post-monsoon months considering the entire span of 50 years (1971–2020) at intervals of 5 years. Splitting the 50-year span to positive, normal and negative IOD years, it was found that the maximum number of cyclones occurred during normal years followed by positive IOD years during post-monsoon months. The impact of IOD events on cyclones in the Indian Ocean basins is very evident. The analysis related to cyclones is conducted in terms of the wave energies generated during the cyclones.

In this study, scatterometer-retrieved wind fields were given as input to wave models during tropical cyclones to simulate cyclonic wave energies. Some locations were chosen along the track of the cyclone to study the magnitude and shape of the energy density spectrum, which calculates the amount of wave energy at different wave frequencies and directions. Model-simulated spectra are in good agreement with the cyclonic wind pattern. Comparison of the model-generated wave spectra with the buoy-observed ones remains as future work as per availability. Spatial wave field plots were generated during the cyclone to analyze the wave patterns. Plots were generated to analyze the intensity and the pattern. The increase in wave energy during extreme events was assessed and quantified for different category cyclones. This study forms a preliminary assessment of cyclone-induced wave energy intensity in the BOB region for an intense positive IOD year and it can be further studied by extension to other regions.

**Author Contributions:** Conceptualization, M.S., S.J. and A.K.; methodology, S.J., M.S. and A.K.; software, M.S. and S.J.; validation, M.S., S.J. and A.K.; formal analysis, M.S., S.J. and A.K.; investigation, M.S., S.J. and A.K.; resources, M.S., S.J. and A.K.; data curation, M.S. and S.J; writing—original draft preparation, M.S. and S.J.; writing—review and editing M.S., S.J. and A.K.; visualization, S.J.; supervision, A.K.; project administration, M.S. and S.J. funding acquisition, M.S., S.J. and A.K. All authors have read and agreed to the published version of the manuscript.

**Funding:** This research received no external funding.

**Data Availability Statement:** Data used in this study are available from the corresponding author on reasonable requests.

**Acknowledgments:** The authors are thankful to Space Application Centre (SAC), Ahmedabad, of the Indian Space Research Organization (ISRO) for the SCATSAT-1 wind datasets (https://www.mosdac.gov.in/ (accessed on 20 November 2019)), NOAA (https://psl.noaa.gov/data/gridded/data.cobe.html (accessed on 29 December 2022)) for the SST datasets and IMD for the cyclone e-atlas (https://rsmcnewdelhi.imd.gov.in/ (accessed on 29 December 2022)).

**Conflicts of Interest:** The authors declare no conflict of interest.

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
