# Peer review of "A Comparison of Wave Spectra during Pre-Monsoon and Post-Monsoon Tropical Cyclones under an Intense Positive IOD Year 2019"

_climate, doi:10.3390/cli11020044_

Round 1

Reviewer 1 Report

A comparison of wave spectra during pre-monsoon and post-2 monsoon tropical cyclones under a positive IOD year

Overall comment: Minor Revision.

 Change the title of paper as ...intense positive IOD year 2019.  I would like to suggest authors …The IOD years are used the SST anomaly of OND or SON season? Clarify it and give Fig of IOD Time series during 1970-2020.

 L11-12 The mode of the coupled oceanic-atmospheric variability comprising the Indian Ocean Dipole (IOD) events are marked by the difference in the sea surface temperature (SST) anomalies  between the tropical western and eastern Indian Ocean. ..Avoid the general statements.

L14  SST anomalies are computed for the pre-monsoon and post-monsoon months for the Bay of Bengal (BOB) and Arabian Sea 14 (AS) from 1971 to 2020, based on the climatology. Author can clear about the how many years used for climatology?

In abstract, authors have to highlight the results instead of methodology and general statements.

L75-79 The 229 unimodal spectra examined were related to hurricanes with central pressures ranging 905 - 985 hPa. There was no conclusion about the directional properties of the spectra in this study. Young (2006) further examined the directional wave spectrum of 9 tropical cyclones along Australia’s North-West coast during 1995-2000, using directional wave buoy data. In all the four quadrants of the hurricane, other than the right rear quadrant, swell dominated spectra were seen.

The sentence not clear and citations are missing. The in manuscript the citations are not in MDPI

Considering observations and model simulations, the dominant wave direction in the front two quadrants radiates out from a region to the right of the hurricane centre. In the right rear quadrant, the waves are wind generated and thus have the same direction. In the left rear quadrant, the wave direction may be from the right of the hurricane center and also may be locally generated. Bi-modal  spectra were obtained for both observed and model simulated wave spectra.

Awful sentences please make sure why authors wrote these lines, avoid general statements and give proper citations.

L176-177 Two regions are chosen in the Indian Ocean, namely BOB (78E – 110E and 25N -10S) and AS (50E – 78E and 25N -10S) and SST data averaged for MAM and OND from 1971 to 2020

-         Degree are missing

the corresponding standardized SST anomalies are computed, authors has to mention the year of climatology used for anomaly.

L 186 CY is the count including depressions, cyclonic storms and se-186 vere cyclonic storms as given by IMD cyclone e-atlas, modify the sentence and what is CY- is it cyclone?

Figure 1 . Scatter plot for all the years (1971-2020) together- what is this caption? Modify it with apprproate sentence. Similarly modify 2-4 Figures as well.

Table 1. Number of cyclones from 1971-2021 at intervals of 5 years, But in table 2016-2020 ...kindly modify accordingly.

Table 2. Number of cyclones IOD year wise...modify this sentence is incomplete. Why Nagative and Nutral IOD gives more cyclones? Give appropriate clarifiacation with citations.

Figure 7. IOD year wise pre and post cyclone frequency in BOB and AS. Modified as “Frequencies of pre and post-monsoon tropical storms during IOD years 1971-2020.

Results sections has modify with approprete discussions. Some of sentences are modified as

L362- 366 The highest waves are observed again on 8th and 9th November but with reduced magnitude. The cyclone had passed as a very severe cyclonic storm through the first location and as a severe cyclonic storm through the second location. Although the magnitude of the wave energy generated reduced, there is no significant change in the spectral shape.

Fig 12 and 13 has to crop for more clarity and captions are modified. I would suggest to authors to take a standard paper of cyclone and change the captions.

 The paper has very good information’s and conclusion part should be compressive and only results have to be highlighted.

Reviewer 2 Report

In this work, the authors compared the wave spectra during monsoon events under India Ocean Dipole from a modelling perspective. The work is surely interesting with extensive literature review, and comprehensive results. The work is of good quality, and can be published in the journal with minor revisions including enriching the methodological details, adding physical interpretation into the discussion of the results along other comments such as:

1.     The authors need to end the introduction with the aim, objective, and scope of the current work, and its novelty compared to the comprehensive literature review presented in the introduction.

2.     A general overview on the basis mathematical formulation of the models are needed to add perspective to the reader.

3.     Lines 176-178, the authors need to support the choice of the regions.

4.     The quality of the plots need to be enhanced, they are very small and hard to read.

5. The majority of the model discussion presents a narrative of the events, rather than adding a physical interpretation of the results, this has to be considered to further enrich the work

Reviewer 3 Report

In this paper, the wave spectral density for cyclones occurring in the pre-monsoon and post-monsoon months is calculated and compared using the SWAN model. I have some main concerns as follows:
1.    Introduction:
It is not clear what is research gap in present studies? Why do you carry out the study? What is your research objective? Please specify the research gap and goal in the introduction.
2.    SWAN model and SCATSAT-1 product and Data and Methodology
“SWAN model and SCATSAT-1 product” Should be part of “Data and Methodology”.
3.    Results and Discussion
(a)    Please use subtitle and subsection to let readers easily understand.
(b)    Please redraw Figures 1 to 4. The y axis should be in the leftmost side. Why does the y-axis show 0,0,0,0,1111…? Very strange. Is there anything wrong?
(c)    In the paper, only two cyclones were selected. Why did the authors the two cyclones? Are they representative? Why not using more cyclone cases to show clearer trend?
(d)    In the SWAN model, there is no comparison of wave heights between the simulation and field observations? How do we confirm the feasibility of the SWAN model?
(e)    In the SWAN model, the daily results mean the average wave height in a one day? There is no description.  

Reviewer 4 Report

Studying wave energy associated with cyclonic storm and relating it to the potential damages caused is a very important and novel topic.

Authors have used gridded SST for 50 years over an extended region to identify the phases of IOD. Authors have collected data of cyclonic storm events over BOB & AS. Initially while reading, it was felt as if authors are going to study wave energy  for storms over BOB & AS in all phases of IOD.

However, it is found that, although authors have provided about some climatic information of formation of Cyclonic storms over BOB & AS during Pre & Post monsoon season. However, such information is not solicited in the present study, because such information are already available in earlier literature, like, Climatology of Energetics of Cyclones over Indian Seas by  Somenath Dutta, Geena Sandhu, Sanjay G Narkhedkar & Sunitha Devi. Authors have studied the wave energy only for two cyclones in only one phase of IOD. Moreover, discussion on wave energy seems to be inadequate.

Authors are suggested to improve the work by following:

1. Categorise  all CS based on basins, seasons and IOD phases.

2. Then run SWAN model for each combination.

3. Composite the wave energy for each combination.

4. Try to relate wave energy with storm surge height, even you prepare a scatter diagram, which will indicate which spectrum is conducive for maximum and minimum height.

Round 2

Reviewer 3 Report

1.     Although the model generated wave heights were under-predicted in comparison with the observation, the authors still can show the cyclonic pattern of the wave fields of measurement and simulated.

2.     In Figures 1 to 4, please move the y axis to the rightmost. The y axis should not cross with the trend line.

3.     In Figures 8 and 12, it is unable to see the velocity vector of the wind. Please redraw the figures and add the vector scale.

4.     Line 131: Figure A? The line and the label of the a-axis should not overlap.

5.     Line 428: Conclusion and caption of Figure 15 are combined.
